# Cytokinin Deficiency Alters Leaf Proteome and Metabolome during Effector-Triggered Immunity in *Arabidopsis thaliana* Plants

**DOI:** 10.3390/plants11162123

**Published:** 2022-08-15

**Authors:** Ling Pan, Miroslav Berka, Martin Černý, Jan Novák, Markéta Luklová, Břetislav Brzobohatý, Iñigo Saiz-Fernández

**Affiliations:** 1College of Forestry, Hainan University, 58 Renmin Avenue, Haikou 570228, China; 2Department of Molecular Biology and Radiobiology, Faculty of AgriSciences, Mendel University in Brno, 61300 Brno, Czech Republic

**Keywords:** *Arabidopsis thaliana*, cytokinin, Flg22 peptide, defence response, metabolome, proteome

## Abstract

The involvement of cytokinins (CK) in biotic stresses has been recognized, while knowledge regarding the effects of CK deficiency on plant response against pathogens is less abundant. Thus, the purpose of this study was to reveal the effects of CK deficiency on proteomics and metabolomic responses of flg22-triggered immunity. We conducted a series of histochemical assays to investigate the activity of the downstream pathways caused by flg22, such as accumulation of ROS, induction of defence genes, and callose deposition, that occurred in *Arabidopsis thaliana* transgenic lines overexpressing the *Hordeum vulgare CKX*2 gene (*HvCKX*2), which are therefore CK-deficient. We also used GC and LC-MS-based technology to quantify variations in stress hormone levels and metabolomic and proteomic responses in flg22-treated *HvCKX2* and wild-type *Arabidopsis* plants. We found that CK deficiency alters the flg22-triggered plant defence response, especially through induction of callose deposition, upregulation of defence response-related proteins, increased amino acid biosynthesis, and regulation of plant photosynthesis. We also indicated that JA might be an important contributor to immune response in plants deficient in CKs. The present study offers new evidence on the fundamental role of endogenous CK in the response to pathogens, as well as the possibility of altering plant biotic tolerance by manipulating CK pools.

## 1. Introduction

The phytohormone cytokinin (CK) is one of the most notable contributors to plant growth and other numerous developmental processes, such as cell growth and differentiation, apical dominance, leaf senescence, and plant stress resistance [1,2,3]. Natural CKs are *N*^6^-substituted adenine derivatives, and their most common forms in plants are *N*^6^-(Δ^2^-isopentenyl) adenines (iPs), *trans*-zeatins (*t*Zs), and *cis*-zeatins (*c*Zs), which are differentially distributed among plant tissues and species [4]. CKs are synthesized by the ISOPENTENYL TRANSFERASE (IPT) and LONELY GUY (LOG) enzymes, whereas the breakdown of CKs and their conjugates occurs mainly through CYTOKININ OXIDASE/DEHYDROGENASE (CKX) enzymes. In *A. thaliana,* a *CKX* gene family (*AtCKX1* to *AtCKX7*) has been identified, and a few studies have elucidated the influence of CK deficiency on several biological processes [5]. For example, overexpression of *AtCKX1* and *AtCKX2* in transgenic plants reduces endogenous cytokinin levels and introduces profound phenotypic changes [6]. A similar reduction in endogenous CK concentration can be achieved using *Arabidopsis* plants overexpressing the *Hordeum vulgare* CKX enzyme (*HvCKX*2) [7]. The employed dexamethasone-inducible expression of barley *HvCKX*2 has two major benefits. First, this isoform was found to be highly active, with broad cytokinin-substrate specificity [8], and its induction effectively depleted the cytokinin pool in transgenic *Arabidopsis* lines [7]. Second, the introduction of an ortholog from a different organism decreases the possibility that the activity of the accumulated protein would be limited by posttranslational modifications. This transgenic line has been reported as successfully employed in many publications, showing its suitability for the analysis of cytokinin-mediated responses to stress [9,10,11]. CK deficiency has been shown to alter sugar biosynthesis, amino acid biosynthesis, and leaf photosynthesis in these *HvCKX*2 transgenic plants [11].

The important regulatory roles for CKs in the enhancement of immune response have already been identified in plants [12]. Emerging evidence has linked increased CK levels to the magnitude of plant immunity, ultimately influencing the outcome of the plant defence response against pathogens [3,13,14]. For instance, application of exogenous CKs lead to an increased defence response through a process either dependent on or independent of salicylic acid (SA) and ethylene (ET) signalling in different plant species [15,16]. Transgenic tomato plants with high levels of CKs show increased production of antioxidants, which appears to counteract the inhibition of photosynthesis, oxidative stress, and stomata closure [17,18]. In *Arabidopsis*, the increase of either exogenous or endogenous CKs leads to increased tolerance to pathogens by triggering JA and SA signalling pathways [19,20]. On the contrary, CK deficiency has been negatively linked to plant tolerance towards pathogens [21,22].

Bacterial flagellin peptide flg22 is an elicitor widely used for triggering early defence response in plants. This effector is recognized by transmembrane pattern recognition receptors and induces a series of immune responses [23,24]. Research performed in the last couple of decades has made it clear that CKs enhance the cellular response to flg22 through a process that relies on SA and MAPK cascade [15] and on the increase of CK-mediated protection strategies in flg22-treated tissues, such as ROS homeostasis and cell wall integrity [21,25,26].

Although much research in recent years has been focused on the elucidation of the indirect and direct roles of endogenous CKs in response to pathogens in *Arabidopsis*, the knowledge on the possible effect of CK deficiency on the regulation of plant defence response is still limited. Furthermore, it is known that CK levels can decrease during several environmental stresses [27] and on aging leaves [28], which can add another level of complexity to the plan–pathogen interactions. Thus, the main purpose of this study is (1) to investigate the effects of CK deficiency on a series of early defence events and on the metabolomic and proteomic response induced by flg22 peptide, and (2) to reveal the mechanisms behind the CK deficiency-mediated regulation of plant defence response. For that, we quantified the effects of CK deficiency on defence-related callose deposition, ROS production, photosynthesis, and CK and JA content in CK-deficient *HvCKX*2 *Arabidopsis* plants treated with flg22 peptide. In addition, we applied integration analysis to highlight the important metabolites and proteins associated with the defence response of *HvCKX2* plants. Overall, this research provides new evidence on the fundamental role of endogenous CK in the response to pathogens.

## 2. Materials and Methods

### 2.1. Plant Material, Growth Conditions, and flg22 Treatment

Seeds of *Arabidopsis thaliana* (L.) dexamethasone (DEX)-inducible line *ProCaMV35S* > GR > *HvCKX2* [7] and wild-type Columbia (Col-0) were grown in soil in a growth chamber with a 16:8 h light:dark regime at 21 °C, 50–60% relative humidity, and 110 µmol m^−2^ sec^−1^ standard light intensity. Plants were activated at 14 DAS (Stage 1.04, four rosette leaves > 1 mm in length with 10 µM DEX dissolved in 5 × 10^−4^% dimethyl sulfoxide (DMSO)). Plants were regularly watered with 1/4 strength Hoagland solution. The third to fifth leaves from the top of the rosette of four-week-old plants were harvested, either whole or in 4 mm disks, and used for subsequent analysis. The flg22 peptide (QRLSTGSRINSAKDDAAGLQIA) was synthesized by Proteogenix (Schiltigheim, France) with >95% purity. The peptide solution was dissolved into sterile, deionized H_2_O to a final concentration of 1 µM.

### 2.2. Histochemical Assays

To detect the dynamic oxidative burst in *Arabidopsis* plants, after collection, leaf discs were submerged overnight in double-distilled water (ddH_2_O). The leaf discs were then placed with adaxial side up in a 96-well illuminometer plate containing 100 µL of the reaction solution (100 µM luminol (Sigma-Aldrich, St. Louis, MO, USA), 10 µg/mL horseradish peroxidase (HRP, Sigma-Aldrich), with or without 1 µM flg22). The light emission was recorded with 1000 ms integration time in 2-min intervals at a period of 60 min (30 cycles) in a microplate reader (Tecan infinite 200-pro; Tecan, Mannedorf, Switzerland). The variable ROS outputs of each treatment were recorded and calculated as the average photon count at each time point (the sum of the photon counts of the leaf discs divided by the discs number, *n* = 6).

For autofluorescence observations of callose deposits, leaves were stained after syringe infiltration with 1 µM flg22 or ddH_2_O (mock) with Sørensen’s phosphate buffer (0.1 M, pH = 8.0) with 0.5% aniline blue solution (CAS number: 28631-66-5, Sigma Aldrich, Darmstadt, Germany). Callose deposits were visualised and photographed with a Olympus Microscope BX61 fluorescence microscope in the field of 40× objective using a DAPI filter equipped with an Olympus DP70 camera. The workflow of image analysis for identification of sites of stained callose was performed using the ImageJ software (https://imagej.nih.gov/ij/ (accessed on 17 December 2020)), followed by “background subtraction”, “noise reduction”, and “image thresholding”. The measure for the level of callose accumulation was presented as the mean grey value obtained from “particle analysis”. At least three images from different areas were taken from each leaf, and five leaves from individual plants were analysed for each treatment.

Stomatal movement detection was performed according to the method described by Eisele et al. [29], with minor modifications. Briefly, leaf discs from *Arabidopsis thaliana* Col-0 and *HvCKX*2 plants were incubated for 2 h in 10 mM MES/KOH buffer (pH 6.15), and the stomatal opening was checked using an Olympus BX61 epifluorescence microscope (Olympus America, Center Valley, PA, USA) set at 20× magnification. After incubation, the leaf discs were immediately stained in 10 µM Rhodamine 6G (Sigma) for 5–10 min, and pictures were taken using the Olympus BX61 epifluorescence microscope. The width and length of the stomata apertures were measured using ImageJ software (https://imagej.nih.gov/ij/ (accessed on 8 July 2020)). Stomata aperture index (SAI) was calculated by dividing the stomata width by the stomata length. Approximately 30 random stomata in four leaf discs from individual plants were used for the calculation of SAI.

### 2.3. Chlorophyll Fluorescence Imaging

Photosynthetic parameters were measured on rosette leaves after syringe infiltration with or without 1 µM flg22 in the specific time range from 0 h to 48 h using Handy FluorCam FC 1000-H (Photon System Instruments Ltd., Brno, Czech Republic). Photosynthetic traits were represented as maximum dark-adapted quantum efficiency F_v_/F_m_, non-photochemical quenching (NPQ), and co-efficient of photochemical quenching of chlorophyll fluorescence (qP). The average value and standard deviation per image were calculated automatically using FluorCam software 7.0. For the imaging chlorophyll fluorescence, at least ten leaves from five individual plants were used for each treatment.

### 2.4. Plant Hormone Analysis

Seven leaves from individual *Arabidopsis* plants were collected per genotype and employed for endogenous plant hormone analysis. The contents of SA and jasmonic acid (JA) were quantified using lipid chromatography coupled with mass spectrometry (LC-MS/MS) as described by [11].

### 2.5. LC-MS-Based Proteomics Assays and Bioinformatic Analysis

Eight rosette leaves from individual plants per genotype were employed for extraction of total proteins and metabolites after infiltration with 1 µM flg22 or ddH_2_O (as mock treatment) for 0 h and 24 h. Total protein extracts were prepared using a combination of phenol/acetone/TCA extraction [30]. Portions of samples corresponding to 5 μg of peptide were analysed by using a nanoflow reverse-phase liquid chromatography-mass spectrometry using a 15 cm C18 Zorbax column (Agilent), a Dionex Ultimate 3000 RSLC nano-UPLC system (Thermo), and a qTOF maXis Impact mass spectrometer (Bruker). The acquired spectra were recalibrated and searched against the Araport *Arabidopsis* protein database by Proteome Discoverer 2.1. The quantitative differences were determined by the spectral counting method, followed by normalisation and *t*-test (compared to the mock-treated roots; *p*-value < 0.05). For selected candidate proteins, the corresponding peptide peak areas were evaluated in Skyline software (version 3.1, MacCossLab Software, Seattle, WA, USA) (https://skyline.gs.washington.edu (accessed on 25 December 2020)). Protein abundance was calculated as the sum of all normalised peptide areas for a given protein. Only significant proteins (|foldchange| ≥ 1.4 and Student’s *t*-test *p*-value < 0.05) were used for biological functional annotation using UniProtKB/Swiss-Prot (https://www.uniprot.org/uniprot/ (accessed on 27 December 2020)), and pathway mapping analysis using the KEGG PATHWAY Database (https://www.genome.jp/kegg/pathway.html (accessed on 27 December 2020)). Cytoscape software (version 3.8.2) was used for analysing and visualising proteomics data.

### 2.6. GC-MS-Based Metabolomics Analyses and Multivariate Statistical Analysis

Polar metabolites were extracted as described by [31] and measured using a Q Exactive GC Orbitrap GC-tandem mass spectrometer and Trace 1300 Gas chromatograph (Thermo Fisher, Waltham, MA, USA). Data were analysed by TraceFinder 4.1 with Deconvolution Plugin 1.4 (Thermo) and searched against NIST2014, GC-Orbitrap Metabolomics library, and the inhouse library. Only metabolites with a score ≥ 75 and ΔRI < 2% were used for the subsequent analysis.

SIMCA 14.1 software (Umetrics, Umea, Sweden) was used for processing multivariate modelling in metabolome profiling. In detail, orthogonal partial least square-discriminant analysis (OPLS-DA) was performed to identify significant metabolites after processing principal component analysis. The cross-validated model explained variance (*Q^2^*Y) and total explained variance (R^2^Y) were represented for evaluating OPLS-DA model quality, including goodness of fit and the overfit of the overall models. The combination of variable importance plot (VIP) larger than 1 and absolute correlation coefficient *p* (corr) larger than 0.5 was used as a cut-off point for identification of important metabolites in OPLS.

The pathway enrichment analysis of metabolomics data was conducted using Metabo Analyst 5.0.

Metabolic network analysis and visualisation of integrated proteomics and metabolomic data were mapped on KEGG pathways using Pathview (https://pathview.uncc.edu/ (accessed on 12 March 2021) and MapMan 3.6.0RC1 based on the above analysis. The abundance changes in metabolome were represented with log2 transformation of fold change values either between flg22-treated plant and mock-treated plants or between transgenic plants and control plants.

### 2.7. Transcriptomic Data Mining

The flg22-responsive genes expressed differentially in the leaves of *Arabidopsis* plants after flg22 treatment (ecotype Columbia) were retrieved from datasets (PRJNA313379). We analysed the differentially expressed gene sets (adjusted *p*-value < 0.05 and |log2FoldChange| > 0.5) using the DESeq2 function in R package (version 4.1.0) [31].

### 2.8. Statistical Analysis

Statistical significance analysis of the means was performed using single-factor ANOVA followed by Duncan’s test and two-factor ANOVA with replication, performed using SPSS statistic v. 25.0 (SPSS Inc., Chicago, IL, USA).

## 3. Results

### 3.1. CK Deficiency Altered flg22-Triggered Early Defence Events

To investigate the role of low CK concentration on early defence events, we measured ROS production and callose accumulation in the leaves of mock-treated and flg22-treated *Arabidopsis* plants (Figure 1). Rapid and significant ROS production was detected in leaves of both *Arabidopsis* genotypes within 1 h after the addition of 1 µM flg22 (Figure 1a,b). The increase in H_2_O_2_ level in *HvCKX2*-overexpressing plants 24 h after flg22 application was almost equal to that in Col_flg22 (Figure 1a,b), although ROS generation was noticeably decreased in *HvCKX2-*overexpressing plants compared to Col_flg22 during the first minutes following flg22 treatment (Figure 1a,b).

In addition, large amounts of callose deposition were found in the leaves of *Arabidopsis* plants after exposure to 1 µM flg22 (Figure 2). This callose accumulation was slightly more pronounced in flg22-treated *HvCKX2* plants than in Col_flg22 plants (Figure 1c). Additionally, the application of 1 µM flg22 caused stomatal closure in *Arabidopsis* leaves within 30 min, yet no significant variations in stomata closure were observed between both *Arabidopsis* genotypes (Figure 1d). Two-factor ANOVA showed that the presence of flg22 was responsible for stomata closure in a CK pool-independent manner.

### 3.2. CK Deficiency Is Involved in the Regulation of Plant Photosynthesis upon Flg22 Peptide Application

To achieve insight into the influence of endogenous CK deficiency on photosynthetic capability in *Arabidopsis* plants, we measured fluorescence induction kinetics on *HvCKX*2-overexpressing plants treated with or without 1 µM flg22. Measurement of chlorophyll fluorescence parameters revealed that the F_v_/F_m_ ratio showed no significant change in flg22-treated *HvCKX*2 plants, which was independent of both CK changes and flg22 treatment (Figure 3a). In contrast, flg22 peptide did cause a significant rise in the quenching coefficient qP in *HvCKX*2 (up to values around 0.35) compared to Col_flg22 (Figure 3b). In addition, we noted that the presence of flg22 sharply altered NPQ values in wild-type plants, while it remained around 1.2 in *HvCKX*2-overexpressing plants, both mock and flg22-treated (Figure 3c). Meanwhile, two-factor ANOVA showed that CK status significantly affected qP and NPQ, while alteration in qP seemed to be independent of the presence of flg22 (Figure 3b,c). Our results underline the involvement of CKs in the adjustment of photosynthetic processes in response to flg22 peptide.

### 3.3. Flg22-Triggered SA Synthesis Was Inhibited by CK Deficiency, While JA Synthesis Seems to Be CK-Independent

To obtain information on whether flg22 peptide alters endogenous levels of SA and JA in CK-deficient *Arabidopsis* plants, we measured the contents of the two stress hormones in the leaves of *HvCKX*2 *Arabidopsis* plants. A significant accumulation of both SA and JA was observed in both *Arabidopsis* genotypes compared to their corresponding mock plants 24 h after flg22 application (Figure 4a,b). It seems that flg22-induced SA-mediated immune response was limited by the CK availability, since SA content in *HvCKX*2_flg22 plants was much lower than in Col_flg22 plants (Figure 4a). Such limitation was not observed in the case of JA, whose levels in flg22-treated *HvCKX*2 plants rose to a similar extent as in Col_flg22 plants, supported by two-factor ANOVA analysis (Figure 4b), suggesting that JA-mediated defence response is independent of CK concentration but is affected by flg22 treatment.

### 3.4. CK Deficiency Specifically Enhanced Amino Acid Biosynthesis and Increased the Sucrose and D-Mannose Concentration in Arabidopsis Leaves upon Flg22 Peptide Application

To assess the effects of CK deficiency on leaf primary metabolism in *Arabidopsis* plants treated with flg22 peptide, we performed a metabolomic profiling of leaf carbohydrates and amino acids. A total of 62 metabolites with significantly affected abundance (*p*-value < 0.05 and VIP > 1) were identified (Appendix A).

A number of sugars (D-fructose and maltose) and amino acids (i.e., L-cysteine, L-leucine, L-histidine, and L-arginine) showed significant upregulation in wild-type plants after flg22 treatment. However, a greater number of sugars and amino acids were affected by flg22 treatment in *HvCKX*2 plants in comparison to the flg22-treated controls (Figure 5). Among them, sucrose, trehalose, and D-mannose significantly accumulated in flg22-treated *HvCKX*2 plants. On the contrary, D-xylose, L-arabinose, L-fucose, and L-rhamnose significantly decreased in the leaves of CK-deficient *Arabidopsis* plants (Figure 5). Similarly, several amino acids showed a significant change in their concentration in *HvCKX*2 plants after application of flg22 peptide, highlighted by an accumulation of L-proline, L-tyrosine, L-alanine, and L-isoleucine (Figure 5). Our results indicate that primary metabolism was largely affected in CK-deficient plants, compared to wild type.

To further investigate how the reduced CK pools affected carbohydrate and amino acid metabolism during defence response in *Arabidopsis* leaves, we distinguished and compared the important metabolites from these pathways in *HvCKX*2 and Col-0 plants after flg22 application (Figure 6; Appendix A). The metabolic maps revealed 15 CK pool-dependent metabolites, including amino acids, which were differentially accumulated in *HvCKX*2 plants compared to their corresponding wild types (Figure 6). In the absence of flg22 treatment, the majority of significantly affected metabolites showed decreased concentration in *HvCKX2* plants compared to wild types, including D-mannose, D-xylose, L-alanine, and succinic acid (Figure 6a). Interestingly, after flg22 treatment, CK deficiency was strongly correlated with upregulation of amino acids, such as L-leucine, L-valine, L-proline, L-methionine, L-isoleucine, L-alanine, L-threonine, and L-arginine, while no significant accumulation of sugars was observed in *HvCKX*2 plants (Figure 6). The reduced CK pool remained negatively correlated with organic acid concentration (i.e., fumarate, 2-oxoglutarate, and succinate; Figure 6b). These results suggest that CK deficiency may stimulate accumulation of important metabolites, especially amino acids, during flg22-induced defence response.

### 3.5. CK Deficiency Altered the Abundance of Proteins Related to Defence Response, Amino Acid Biosynthesis, and Cell Wall Integrity

To reveal how CK-deficient *Arabidopsis* plants react to bacterial flg22 peptides at the proteomic level, comparative analyses were performed on proteomes of *HvCKX2* plants and wild-type plants subjected to flg22 peptides. We quantified 137 significantly affected proteins in both *Arabidopsis* genotypes (Appendix A).

To clearly determine flg22-induced gene expression in *HvCKX2* plants, we performed transcriptomic data mining to identify proteins encoded by flg22-induced genes in the controls (Appendix A). A total of 42 flg22-reponsive proteins showed high abundance in processes related to defence response (Glutathione *S*-transferase F7 (GSTF7), Glutathione *S*-transferase F6 (GSTF6), Thioredoxin H5 (TRX5), Trypsin inhibitor1 (TI1), LRR receptor-like serine/threonine-protein kinase (IOS1), and Leucine-rich repeat serine/threonine-protein kinase 1 (LRK1)), lignin biosynthesis (i.e., Cinnamyl alcohol dehydrogenase 5 (CAD5), Flavone 3′-*O*-methyltransferase (OMT1), and Cinnamyl alcohol dehydrogenase 8 (ELI3-2)), and peroxidase activity (i.e., Peroxidase 34 (PRXCB), Peroxidase 71 (At5g64120), and Peroxidase 21 (At2g37130)) in both *HvCKX2* and wild-type plants treated with 1 µM flg22 (Figure 7a,b). In contrast, treatment of leaves with flg22 distinctly lowered the abundance of Glycolate oxidase 2 (GOX2) and Glycolate oxidase 1 (GOX1), which are involved in photorespiration, in both *Arabidopsis* genotypes (Figure 7b).

Furthermore, we also detected an increase in the abundance of stress-related proteins in CK-deficient plants exposed to flg22, relative to Col_flg22 plants (Appendix A). Protein integration networks were created to explore the functional interactions of the proteins in various biological processes. Unlike similar patterns observed among abundances of flg22-responsive proteins, the expression of CK pool-dependent proteins was largely distinct (Figure 7c). The abundance of proteins regulating the breakdown of cell walls (i.e., Expansin-A1 (EXPA1), Expansin-A8 (EXPA8), and Alpha-xylosidase 1 (XYL1)) and starch catabolic and biosynthesis processes (i.e., Beta-amylase (CT-BMY), Pullulanase 1 (LDA), and Starch synthase 1 (SS1)) was strongly suppressed (Figure 7c). Additionally, upregulation of two defence proteins (Kunitz trypsin inhibitor 2 (KTI) and Glutathione *S*-transferase DHAR1 (DHAR1)) and two antioxidants that mediate ROS scavenging (Superoxide dismutase [Cu-Zn] 1 (CSD1) and Superoxide dismutase [Cu-Zn] 2 (CSD2)) increased significantly (two times or more) in leaves of *HvCKX2* plants. In contrast, several proteins related to the biosynthetic pathways of L-valine, L-isoleucine, and L-lysine showed slightly decreased abundance in *HvCKX2* plants (Figure 7c). These results indicated that CK pool-dependent proteins are important to defence response, balance of amino acid metabolism, and maintenance of cell wall integrity.

## 4. Discussion

The rapid and strong production of ROS in the apoplast is one of the earliest detectable events that occur in response to biotic stress and is part of the hypersensitive response employed by plants to fight many biotrophic and hemi-biotrophic pathogens [32,33]. This ROS outburst is believed to be mediated by SA and repressed by JA [24]. To this regard, our results showed that *HvCKX*2 plants were not able to increase their SA levels to the same extent as wild types when coping with the application of flg22 but increased their JA content (Figure 4), which is in line with the suggested antagonistic nature of SA and JA signalling pathways [24]. The observed increase in JA could have been a response to the inability of *HvCKX*2 plants to achieve the proper SA levels, thus requiring the strengthening of alternative defence mechanisms, as suggested by Muñoz-Espinoza et al. [34] and Brouwer et al. [35]. In fact, some studies have already shown an increase in JA concentration [36] and signalling [37] in SA-deficient *Arabidopsis* mutants. However, despite having a lower SA concentration than their Col-0 counterparts, *HvCKX*2 plants were able increase their ROS levels after flg22 application, albeit to a lesser extent than wild types (Figure 1a). This data supports the correlation between SA levels and ROS outburst, but also proves that, as suggested by Dewdney et al. [38], low levels of SA are enough to produce the hypersensitive response, at least to a certain extent. The reduction of H_2_O_2_ in *HvCKX*2-overexpressing plants further supports the idea that proper CK and SA levels are required to display a full hypersensitive response against pathogen attack.

In keeping with this, the abundance of several genes encoding for antioxidant enzymes and non-enzymatic antioxidants involved in the maintenance of low levels of ROS were increased in *HvCKX2* plants (Figure 7). Namely, ROS scavenging enzymes involved in the removal of the oxidative damage-causing H_2_O_2_, such as peroxidase 21 (At2g37130), peroxidase 71 (At5g64120), and peroxidase 34 (PRXCB), were found to increase upon flg22 application. Glutathione is an antioxidant crucial to maintain intracellular redox homeostasis [39]. Several glutathione-related enzymes and glutathione transferases (i.e., GSTF6, GSTF7, and DHAR1) were found to be highly abundant in *HvCKX2* plants (Figure 7a), confirming the role of glutathione in the regulation of ROS mitigation in plants with altered CK content in response to pathogen attack [40]. Furthermore, L-proline, L-methionine, L-tyrosine, and L-lysine could have also made a substantial contribution to mitigation of ROS in *HvCKX2* plants, thanks to their strong antioxidant protecting properties [41]. In addition to flg22-induced antioxidant production, CK deficiency also promoted the abundance of the antioxidants CSD1 and CSD2, which encode for enzymatic antioxidant-Cu/Zn-SODs that are associated with oxidative stress tolerance [42].

SA and ROS outbursts are believed to be among the necessary components for callose deposition in response to various biotic stresses [24,43]. Interestingly, *HvCKX*2 plants were able to achieve a slight increase in callose deposition after treatment with flg22 (Figure 1c and Figure 2), despite showing a decreased SA content and ROS outburst compared to wild types (Figure 1a and Figure 4a). To this regard, CK-deficient plants have been shown to able to achieve considerable amounts of callose deposition, not much smaller than wild types [20], and Yi et al. [24] suggested that proper callose deposition can be achieved under decreased SA and ROS levels. In addition, Vojta et al. [44] showed that *CKX-*overexpressing plants had upregulated expression of 50% of the genes related to defence response by callose deposition. This shows that the relationship between CKs, SA, ROS, and callose deposition deserves to be further investigated.

Callose accumulation in the cell walls of *Arabidopsis* plants is believed to be necessary for plant defence response [45]. This callose deposition can be beneficial, as it improves cell wall rigidity and indicates activity of plant immunity [46,47]. Cell wall rigidity can also be achieved by lignification processes. To this regard, the proteomic data shows that the abundance of Flavone 3′-*O*-methyltransferase 1 (OMT1), Cinnamyl alcohol dehydrogenase 5 (CAD5), and Cinnamyl alcohol dehydrogenase 8 (ELI3-2), required for lignin formation, was increased during plant response to flg22 peptide [48], as suggested by our study. More importantly, *HvCKX2* plants showed a steep decrease in the abundance of proteins involved in degradation of cell walls, such as EXPA8, EXPA1, and LP1. This result confirmed the importance of CK for modulating cell wall integrity in *Arabidopsis*, in line with previous studies [49].

Even though previous experiments have shown that decreased CK pools can lead to a decrease in leaf sugars in *Arabidopsis* during normal growth conditions [11]; here, a significant sugar accumulation was observed in *HvCKX2* plants during the flg22-induced defence response (Figure 5). This increase in sugar content could have been a consequence of decreased starch synthesis [50,51], which is regulated by CK pools to some extent [52]. Accordingly, we observed the repression of enzymes related to starch biosynthesis, such as CT-BMY, LDA, and SS1 (Figure 7c). The resources that were redirected away from starch synthesis, presumably towards metabolic pathways downstream of glycolysis, could have been used to fuel the observed increase in amino acid biosynthesis in *HvCKX*2 plants. Additionally, the decrease in photorespiration perhaps contributed to the increased sugar content in *HvCKX*2 plants in the presence of flg22 treatment (Figure 3 and Figure 5), since it is considered a sugar-consuming process [53]. To this regard, having been long recognized as participators in the photorespiratory pathway in *Arabidopsis* plants, GOX1 and GOX2 were less abundant in *HvCKX2* plants, indicating that photorespiration could have been repressed in *HvCKX2* plants [54]. It should be noted that along with ROS inhibition, CK deficiency might have also suppressed the accumulation of O_2_ that binds to ribulose-1,5-bisphosphate in the Calvin cycle instead of CO_2_, thus decreasing photorespiration [53]. Similarly, qP, a major factor representing the ability of leaves to convert light energy to chemical energy [55], rose in *HvCKX2* plants exposed to flg22 peptide, while it did not increase in Col-0 plants, which also displayed a lower basal value (Figure 3b). This illustrates how under flg22 treatment, leaves with lower CK content tend to absorb more light energy that can later be used to drive photosynthesis. Part of this absorbed energy can be dissipated as heat, in a process called non-photochemical quenching (NPQ), as an attempt to reduce photodamage during unfavorable conditions [56]. This chlorophyll fluorescence parameter has been found to be rapidly reduced by PAMPs [57]. Interestingly, flg22 application seemingly did not cause any increase in the excitation energy transfer and dissipation in *HvCKX2* plants (Figure 3c), which was probably because the exposure to the flg22 peptide forced stomatal closure, inhibiting CO_2_ fixation (Figure 1d). This could have led to a suppression of electron transport though photosystem II (PSII), yet increased the cyclic electron flow involving only PSI in *HvCKX2* plants, which helped plants avoid the increase in NPQ and photodamage.

## 5. Conclusions

We have provided a comprehensive study of the effects of CK deficiency on plant immune response in *Arabidopsis*. Our data show that CK deficiency alters the flg22-triggered plant defence response through modifications in the amino acid and sugar biosynthesis and plant photosynthesis. Despite showing lower SA content and a decreased ROS outburst, CK deficiency does not prevent plants from achieving the required amount of callose deposition. In addition, alternative defence mechanisms seem to be activated in CK-deficient plants, highlighted by the increase in defence-related proteins and JA content, and the decrease in photorespiration. This study highlights the alterations in the early immune response of *Arabidopsis* plants with decreased CK pools and thus hints at the possibility of modifying plant defence response by manipulating their CK content.

## Figures and Tables

**Figure 1 plants-11-02123-f001:**
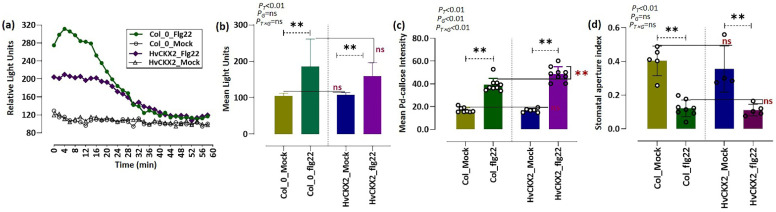
(**a**,**b**) Representative flg22-induced ROS outburst in *Arabidopsis* plants. The mean photon count is represented as relative light units during the first 60 min (**a**) and 24 h (**b**) after treatment with 1 μM flg22 or ddH_2_O (mock). (**c**) The intensity of flg22-triggered callose deposition in the leaves of *Arabidopsis* Col-0 and *HvCKX2* genotypes 24 h after treatment with 1 μM flg22 or ddH_2_O (mock). (**d**) Flg22-triggered stomata closure in *Arabidopsis* leaves, presented as stomatal aperture index in the leaves of *Arabidopsis* Col-0 and *HvCKX2* genotypes 24 h after treatment with 1 μM flg22 or ddH_2_O (mock). Asterisks indicate statistical differences between the indicated treatments, calculated using two-sided Student’s *t*-test (** significant at *p* < 0.001; ns, not significant). Probability value (*p*) indicates the results of two-factor ANOVA (letter ‘T’ represents treatment (Flg22 and Mock); letter ‘G’ represents genotype (*HvCKX2* and Col_0); ‘T × G’ represents interaction between genotype and treatment).

**Figure 2 plants-11-02123-f002:**
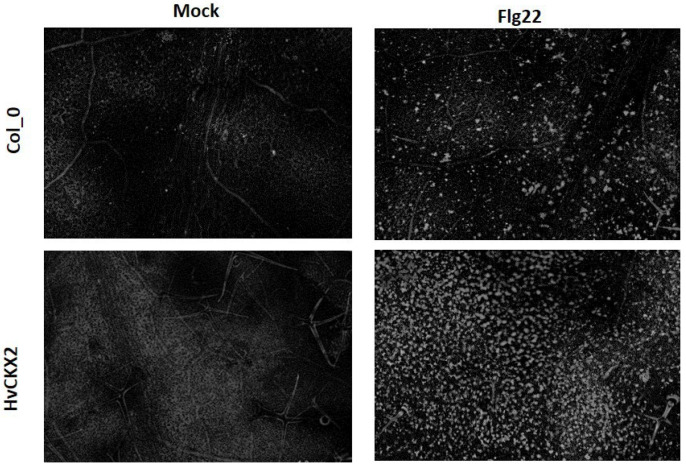
flg22-triggered callose deposition in the leaves of *Arabidopsis* Col-0 and *HvCKX2* genotypes 24 h after treatment with 1 μM flg22 or ddH_2_O (mock).

**Figure 3 plants-11-02123-f003:**
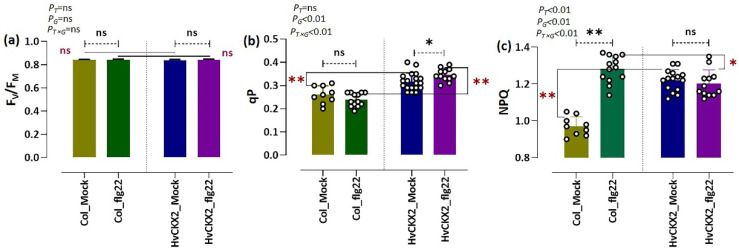
(**a**–**c**) Photosynthetic parameters of *Arabidopsis* plants treated with or without 1 µM flg22, represented in terms of maximum PSII quantum yield (F_V_/F_M_), coefficient of photochemical quenching of chlorophyll fluorescence (qP), and non-photochemical quenching (NPQ). Bars represent the mean of at least ten biological replicates. Error bars indicate standard deviation. Asterisks represent statistical differences between flg22-treated *Arabidopsis* plants and their mock plants (highlighted with black-coloured asterisk) or between Col-0 and *HvCKX*2 plants within the same treatment (highlighted with red-coloured asterisk) (* significant at *p* < 0.05; ** significant at *p* < 0.001; ns, not significant; two-sided Student’s *t*-test). Probability value (*p*) indicates the results of two-factor ANOVA (letter ‘T’ represents treatment (Flg22 and Mock); letter ‘G’ represents genotype (*HvCKX2* and Col_0); ’T × G’ represents interaction between genotype and treatment).

**Figure 4 plants-11-02123-f004:**
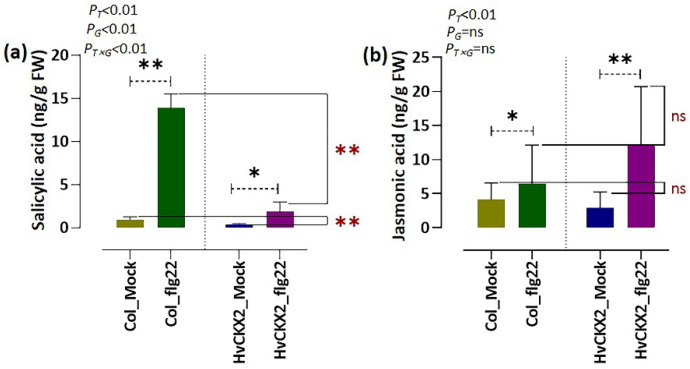
Concentration of (**a**) salicylic acid (SA) and (**b**) jasmonic acid (JA) in the leaves of *Arabidopsis* Col-0 and *HvCKX2* genotypes 24 h after treatment with 1 μM flg22 or ddH_2_O (mock). Bars represent the mean of seven biological replicates. Black asterisks indicate significant differences between flg22-treated *Arabidopsis* plants and their corresponding control plants within the same genotype. Red asterisks indicate significant differences between *HvCKX2 Arabidopsis* plants and wild-type plants within the same treatment (* significant at *p <* 0.05; ** significant at *p <* 0.001; ns, not significant; two-sided Student’s *t*-test). Probability value (*p*) indicates results of two-factor ANOVA (letter ‘T’ represents treatment (Flg22 and Mock); letter ‘G’ represents genotype (*HvCKX2* and Col_0); ’T × G’ represents interaction between genotype and treatment).

**Figure 5 plants-11-02123-f005:**
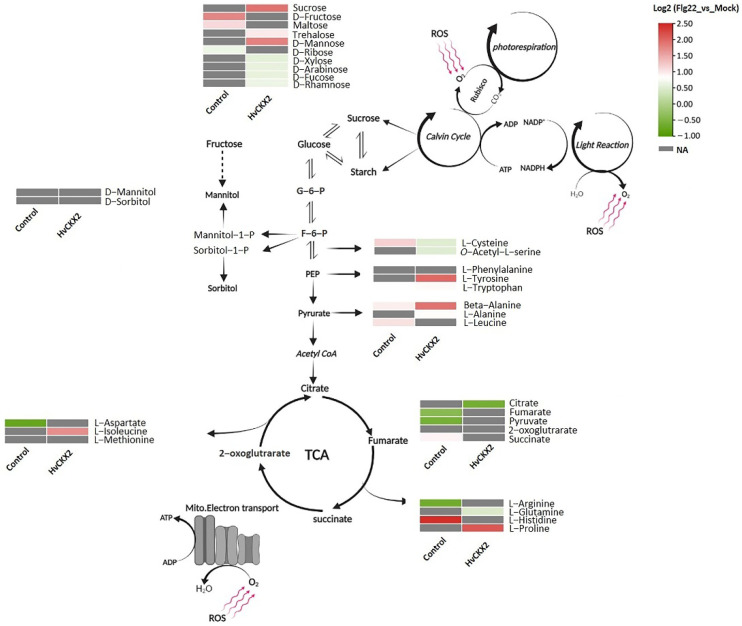
Diagram highlighting the metabolic changes observed in Arabidopsis *HvCKX*2 and Col-0 plants 24 h after treatment with 1 µM flg22. The heatmaps show the log2-fold changes in the concentration of each metabolite between control and flg22-treated plants of both genotypes. Metabolites with no statistically significant changes are shown in grey (NA—not affected).

**Figure 6 plants-11-02123-f006:**
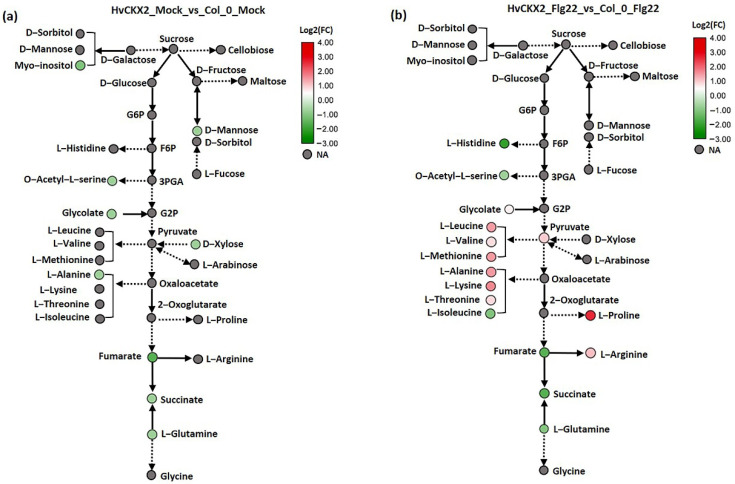
(**a**,**b**) Network-integrated visualization of altered metabolites in *HvCKX2* plants treated without or with flg22 relative to the corresponding wild types. The coloured circles highlight CK pool-sensitive metabolites, according to the metabolome comparative analysis of *HvCKX*2 and wild-type genotypes. Metabolites with no statistically significant changes are shown in grey (NA—not affected).

**Figure 7 plants-11-02123-f007:**
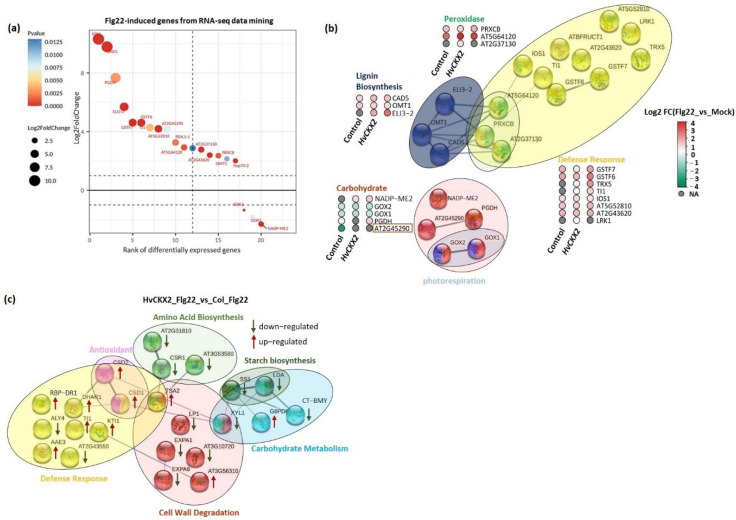
(**a**) Expression of flg22-induced genes encoding the flg22-responsive proteins obtained from RNA data mining. (**b**,**c**) Protein–protein associations produced with STRING database which present the functional partnerships between significant flg22-responsive proteins and CK pool-dependent proteins involved in plant defence responses. The heatmaps represent increased (red) or decreased (green) abundance of specific proteins in *HvCKX2* plants compared with their corresponding controls.

## Data Availability

The manuscript includes all the data.

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
