# Peer review of "Cytokinin Deficiency Alters Leaf Proteome and Metabolome during Effector-Triggered Immunity in Arabidopsis thaliana Plants"

_plants, 2022, doi:10.3390/plants11162123_

Round 1
Reviewer 1 Report
Dear Authors,
I have reviewed the manuscript "Proteome and metabolome dynamics underlying attenuated effector-triggered immunity response in cytokinin deficient Arabidopsis thaliana plants". I find that your manuscript currently looks like a half-product which is far from being acceptable for publication in Plants, and an array of corrections will be necessary before it can be reconsidered for publication.
For my detailed comments on your manuscript and suggestions for improvement, please consult the following:
- Title – The title of your manuscript is too complicated and confusing. Please replace "attenuated effector-triggered immunity response" by clearer and more readily understandable wording
- Abstract
o line 18: HvCKX2 is a gene originating from barley. This should be stated in the Abstract.
o line 19-21: This sentence is not sufficiently specific. It is unclear in what way the defence response-related proteins, synthesis of amino acids, and regulation of photosynthesis, are affected by cytokinin deficiency.
- Introduction
o line 49-52: This part of the text is poorly written. Exogenous and endogenous CKs lead to enhanced susceptibility? Exogenous CKs cannot lead to enhanced susceptibility. Exogenous CKs are taken up by plant tissues and become endogenous, before they can have any kind of effect on plants. I believe that you meant to say "THE INCREASE of either exogenous or endogenous CKs..."
o line 51-52: The next sentence "On the contrary, ..." is in complete, illogical contradiction with the previous one. Please provide context for this sentence.
o line 53-55: "The defence response in plants is recognized by transmembrane pattern recognition receptors"??? Please re-write the entire sentence.
- Aim of the research – the aim of your research is not appropriately framed, which makes the context of your research and the relevance of your results unclear:
o please provide a rationale as to why would the plant immune response to flg22 be of particular interest in cytokinin-deficient plants. Why would we want this kind of knowledge, i.e., what is its scientific significance?
o please provide a justification for the use of a gene from barley for the transformation of Arabidopsis. Why did you use a gene from barley when seven genes from Arabidopsis are characterised and available?
o line 65-71: this part of text sounds like a conclusion to the article, rather than the aim of the research. It should be shortened, leaving out the obtained results and conclusions drawn from the research.
- Material & Methods
o Section 2.1: I do not understand the connection between plant material and peptide synthesis. These two parts definitely do not belong in the same subsection.
o line 77: please provide a reference for the HvCKX2 transgenic line used in your research
o line 80: The reference "Boyes et al." is missing from the References list
o line 109: The reference "Eisele et al." is missing from the References list
o line 108-114: This entire part is written in a very confusing manner and should be re-written from scratch.
- Results – a general remark for the Results is that they are presented in a very confusing manner (esp. those in Figures 4-6) which makes it difficult for the reader to discern what are the effects of cytokinin deficiency alone, what are the effects of flg22 alone, and what are the effects of cytokinin deficiency on the plant response to flg22. Since the complex interaction between cytokinin deficiency and the plant response to flg22 seems to be the central topic of your manuscript, the aim of your research is not met until you clearly point out the specific effects of the interaction between these two factors in your paper.
o Fig. 1b: Which time point is shown in Figure 1b? Please specify in the figure caption.
o line 199: Are any photographs of the callose depositions in the Arabidopsis plants available? Please include them if so
o line 210: after treatment WITH 1 μM flg22
o line 241: at what time point were the levels of SA and JA measured in the leaves?
o line 248: Is it CK-independent for sure? I suggest that for both SA and JA, you should calculate and present the fold-increase (with the whole statistical analysis) of the endogenous levels upon the flg22 treatment, beside absolute values. Looking at Figure 3b, to me it seems that the fold-increase of JA upon exposure to flg22, might be importantly greater in HvCKX2 plants than in Col0. Could it be that the JA-dependent pathway is switched on more efficiently in the cytokinin-deficient plants as a compensation for their inability to efficiently switch on the SA-dependent pathway?
o line 257: sugars are missing from the subsection title
o lines 263-271: It is completely unclear from this paragraph, as well as from the accompanying Fig. 4, whether this pattern of flg22-induced changes in the sugar metabolome is different between wild-type and cytokinin-deficient plants. Even though it might make the text look excessively long and cumbersome, it is necessary to be clear at this point because these results appear to be of central importance to your manuscript. Please re-write this whole paragraph, and make sure that you:
ï‚§ first state the differences between wild-type and cytokinin-deficient plants in control conditions;
ï‚§ then proceed to describe the changes brought about by the exposure to flg22 in wild-type plants;
ï‚§ then, finally (and most importantly), point out the important differences between the flg22-induced changes in wild-type plants, and the flg22-induced changes in cytokinin-deficient plants. Not the differences between the treated plants; we need to see the differences between the changes that occur in the two genotypes.
ï‚§ This way you will be able to point out clearly how cytokinin deficiency specifically affects the responses of Arabidopsis plants to the exposure to flg22.
o Fig. 4: The conceptualization for this figure is excellent, but the figure itself is flawed. As far as I understand the colour scale, the various nuances of red-to-green stand for how much a certain metabolite is altered by the exposure to flg22 in wild-type, or in HvCKX2 plants. However, the dark grey colour stands for "NA" (Not applicable? Not available?) I would not even notice if the "not applicable" colour stood for one or two metabolites, but it stands there virtually for at least one of the genotypes next to every metabolite. Please provide an explanation as to what "NA" stands for, and re-check your data to make the diagram informative as appropriate.
o line 276: "schematic diagram with heatmaps highlighted metabolic changes" – this figure caption is a total mess. Beside being grammatically incorrect it is abundant with superfluous words (i.e., every diagram is always necessarily schematic). You might want to change this into something like "Heatmap representation of metabolic changes in flg22-responsive..."
o line 284-286: This sentence is wrong on at least two levels. It sounds like you are introducing the differences between complete metabolomics of untreated Col0 and HvCKX2 plants, whereas you are actually about to introduce the specific differences in amino acid content, similarly like you did for the sugars in the previous paragraph. Kindly revise.
o line 287: "plant response", not "plant's response"
o the entire paragraph 284-297: for the sake of clarity of your results, please try to stick to the same order as for the paragraph 263-271:
ï‚§ 1) differences between wild-type and cytokinin-deficient plants in control conditions;
ï‚§ 2) most important changes induced by flg22 in the metabolome of wild-type plants;
ï‚§ 3) most important differences between what is induced in wild-type plants, and what is induced in cytokinin-deficient plants, by flg22
o Fig. 5 is far less understandable than Fig. 4 in terms of what is actually shown in the figure. Whereas Fig. 4 (supposedly?) shows how much is each metabolite altered in wild-type, or in HvCKX2, by the exposure to flg22, Fig. 5 shows just only one colour for each metabolite. Why?? What does this colour represent?
ï‚§ upregulation by flg22 in HvCKX2 plants?
ï‚§ upregulation by flg22 in control plants?
ï‚§ the ratio between these two upregulations?
ï‚§ Please thoroughly revise Fig. 5 and build it on the same logic that you used for Fig. 4.
ï‚§ The figure caption for Fig. 5 needs to be completely re-written in order to explain what is actually shown in the figure. You should not leave it up to the reader to guess what it is that you wanted to show in the diagram.
o line 303: You cannot say that the plants "deal" with the peptide. Rephrase to "react to".
o lines 310-314, 322-325: The abbreviated names of each protein should be fully spelled at first mention.
o line 326: "several proteins in the biosynthetic pathways..." – please correct to "several enzymes" if that is what you meant
o Figure 6:
ï‚§ 6b, text within the figure itself – here you are spelling "defense" in American English spelling, whereas you used the British spelling "defence" everywhere else in the manuscript. Please keep the consistency.
ï‚§ 6d – Figure caption mentions four subfigures (6a, b, c, d) whereas the figure contains only a, b, and c.
ï‚§ Figure caption – revise the figure caption and make it more detailed to point out what is exactly shown in each of the subfigures. Please write a distinct caption for each subfigure. The comment applies to figures 1-5 as well. 4
o line 335: Exactly at a point where one would expect the word ("green") in the brackets, suddenly a reference which is missing from the reference list ("Mach et al."), appears in the brackets, apparently for no reason. Please double-check and revise if necessary.
- Discussion – The results are not properly discussed at all and the entire Discussion section should be re-written from scratch. Among other things, it should be pointed out and discussed whether the observed differences between HvCKX2 and wild-type Arabidopsis plants in response to flg22 can be interpreted as superior or inferior adaptability of HvCKX2 plants as a result of cytokinin deficiency. Appropriate Conclusions to your work should be written accordingly. Detailed comments on the Discussion follow:
o The entire discussion calls for thorough commentary on the interpretation of the adaptive significance of the differentially expressed genes, proteins, and metabolites. Some genes, proteins, or metabolites, that protect the plant tissues against oxidative stress, might be more efficiently upregulated in the cytokinin-deficient plants than in the Col0 plants. Does this necessarily mean that the cytokinin-deficient plants are fitter for the mitigation of stress than the Col0 plants? Or, can it be hypothesised that the protective mechanisms that are more upregulated in HvCKX2 than in Col0, are actually less efficient, "backup" strategies for stress mitigation, that need to be switched on for the plants to respond to stress when their primary stress response mechanisms are insufficient (i.e., due to cytokinin deficiency)?
ï‚§ Please discuss more on this issue and provide evidence that would support the "greater adaptivity" or "lesser adaptivity" of the cytokinin-deficient plants for the major differences between the HvCKX2 and Col0 plants in response to stress.
ï‚§ Try to build your entire discussion around the "greater adaptivity" or "lesser adaptivity" paradigm, instead of discussing every trait that you measured one by one, without a logical thread that connects them all.
o line 356: the names of the proteins need to be fully spelled at first mention
o line 388-390: Why is it in contradiction? If cytokinin deficiency leads to lowering the sugar levels in the leaves in normal conditions, does that necessarily mean that the sugars will be unable to accumulate in response to stress? I do not think that these results are contradictory per se.
o line 396-399: Is this a difference from the Col0 plants? If so, please provide brief commentary
o Entire passages within the current version of Discussion are unsound, unfounded, or devoid of meaning. These parts should be thoroughly re-written, or even deleted from the Discussion:
ï‚§ line 343-345: This sentence is completely incorrect. The levels of SA are increased in both HvCKX2 and Col0 plants when they are exposed to flg22. Furthermore, the levels of SA and JA are not shown in Figure 1 but in Figure 3. Please read your own text after you write it, and aim for your sentences to be specific, and technically correct.
ï‚§ line 347-348: Whatever you wanted to say in this sentence, is completely incorrect. SA and JA are not "antagonistically regulated".
ï‚§ line 365: "were affected" – How affected? "+" or "-" ?
ï‚§ line 366-373: This part of text is written in very confusing manner, and written like this, it definitely "does not show" anything.
- Conclusion – please make the Conclusions section separate from the Discussion, and re-write it from scratch after you have thoroughly re-conceived your Discussion around the hypothesis of adaptive or counter-adaptive meaning of the observed differences between HvCKX2 and Col0 plants in response to the flg22 peptide.
- Inconsistent usage of italic letters, capital letters, superscript, and other – Throughout your manuscript, latin names (such as Arabidopsis), gene names (such as HvCKX2), statistically-related terms (such as p-value or Student's t-test), conformation of cytokinins (cis-zeatin and trans-zeatin) which should be written in italic, are left in plain text 9 times out of 10. Furthermore, the names of sugars and amino acids are written randomly in lowercase letters or capitalised, and with or without the designation of molecular conformation (L- or D-) without any criterium whatsoever (e.g., in line 290-292, you write: "L-leucine, L-valine, proline, Methionine and L-isoleucine, alanine, threonine, and L-arginine"). This makes the text of your manuscript look untidy and messy, which is a bit direspectful to the readers. Please take care of that.
Author Response
Response to Reviewer 1 Comments
Point 1: - Title – The title of your manuscript is too complicated and confusing. Please replace "attenuated effector-triggered immunity response" by clearer and more readily understandable wording
Response 1: We have replaced the previous Title with New Title ‘Cytokinin deficiency alters leaf proteome and metabolome during effector-triggered immunity in Arabidopsis thaliana plants’.
Point 2: line 18: HvCKX2 is a gene originating from barley. This should be stated in the Abstract.
Response 2: We have now mentioned this in the abstract.
Point 3: line 19-21: This sentence is not sufficiently specific. It is unclear in what way the defence response-related proteins, synthesis of amino acids, and regulation of photosynthesis, are affected by cytokinin deficiency.
Response 3: We have modified the sentence to be a bit more specific. However, we cannot go into much detail in the abstract.
Point 4: line 49-52: This part of the text is poorly written. Exogenous and endogenous CKs lead to enhanced susceptibility? Exogenous CKs cannot lead to enhanced susceptibility. Exogenous CKs are taken up by plant tissues and become endogenous, before they can have any kind of effect on plants. I believe that you meant to say "THE INCREASE of either exogenous or endogenous CKs..."
Response 4: The reviewer is right. We have corrected the sentence accordingly.
Point 5: line 51-52: The next sentence "On the contrary, ..." is in complete, illogical contradiction with the previous one. Please provide context for this sentence.
Response 5: The previous sentence was incorrect. It should have said that increase in CKs leads to an increase in plant tolerance (which has now been corrected). The following sentence (“On the contrary…”) thus refers to CK deficiency decreasing plant tolerance.
Point 6: line 53-55: "The defence response in plants is recognized by transmembrane pattern recognition receptors"??? Please re-write the entire sentence.
Response 6: The elicitor flg22 is the one being recognized by transmembrane factors. The reviewer is right; the sentence was not. We have rephrased it.
Point 7: - Aim of the research – the aim of your research is not appropriately framed, which makes the context of your research and the relevance of your results unclear: please provide a rationale as to why would the plant immune response to flg22 be of particular interest in cytokinin-deficient plants. Why would we want this kind of knowledge, i.e., what is its scientific significance?
Response 7: CK content is known to decrease during certain environmental stresses and at certain developmental stages (such as in old leaves). If plants or tissues in any of these situations face the attack of a pathogen, they will have a reduced CK pool compared to a plant/tissue growing in more standard conditions. This adds complexity to the plant-pathogen interaction and will affect plant defence response. Therefore, the study of CK-deficient plant response to flg22 could provide useful information about how plants growing in conditions that lower their CK pools would react to a pathogen attack. We have now mentioned this in the introduction.
Point 8: please provide a justification for the use of a gene from barley for the transformation of Arabidopsis. Why did you use a gene from barley when seven genes from Arabidopsis are characterised and available?
Response 8: The gene from barley is less susceptible to silencing or modification within Arabidopsis plants and has been shown to be stable and very effectively degrade endogenous CKs in Arabidopsis transgenic lines.
Point 9: line 65-71: this part of text sounds like a conclusion to the article, rather than the aim of the research. It should be shortened, leaving out the obtained results and conclusions drawn from the research.
Response 9: We have rephrased and shortened this part of the text.
Point 10: Section 2.1: I do not understand the connection between plant material and peptide synthesis. These two parts definitely do not belong in the same subsection.
Response 10: Our intention was to include in this subsection all the information about plant material, growth conditions, and application of the flg22 peptide. Basically, everything that did not involve the obtaining of data. With that mindset, peptide synthesis needed to be included in that subsection. We have now renamed the subsection “2.1. Plant Material, growth conditions and flg22 treatment”
Point 11: line 77: please provide a reference for the HvCKX2 transgenic line used in your research.
Response 11: We have included the reference for the HvCKX2 transgenic lines.
Point 12: line 80: The reference "Boyes et al." is missing from the References list
Response 12: We have included this citation in the list.
Point 13: line 109: The reference "Eisele et al." is missing from the References list
Response 13: We have included this citation in the list.
Point 14: line 108-114: This entire part is written in a very confusing manner and should be re-written from scratch.
Response 14: We have rewritten this part of the text.
Point 15: Fig. 1b: Which time point is shown in Figure 1b? Please specify in the figure caption.
Response 15: It shows data from 24 hours after treatment. This has now been included in the caption.
Point 16: line 199: Are any photographs of the callose depositions in the Arabidopsis plants available? Please include them if so
Response 16: This has been added.
Point 17: line 210: after treatment WITH 1 μM flg22
Response 17: This has been corrected.
Point 18: line 241: at what time point were the levels of SA and JA measured in the leaves?
Response 18: It was measured 24 hours after treatment. This has now been specified both in the text and in the Figure 3 caption.
Point 19: line 248: Is it CK-independent for sure? I suggest that for both SA and JA, you should calculate and present the fold-increase (with the whole statistical analysis) of the endogenous levels upon the flg22 treatment, beside absolute values. Looking at Figure 3b, to me it seems that the fold-increase of JA upon exposure to flg22, might be importantly greater in HvCKX2 plants than in Col0. Could it be that the JA-dependent pathway is switched on more efficiently in the cytokinin-deficient plants as a compensation for their inability to efficiently switch on the SA-dependent pathway?
Response 19: The increase in JA content was indeed bigger in HvCKX2 plants after flg22 application, even though the differences in the concentration between HvCKX2_flg22 and Col-0_flg22 were not statistically significant. We have mentioned in the discussion the possibility of CK-deficient plants may need to switch towards a JA-mediated defence response. However, our data shows that plants were able to increase JA concentration even when the CK concentration was low, suggesting that this pathway is CK-independent.
Point 20: line 257: sugars are missing from the subsection title
Response 20: We have now mentioned sugars in the subsection title.
Point 21: lines 263-271: It is completely unclear from this paragraph, as well as from the accompanying Fig. 4, whether this pattern of flg22-induced changes in the sugar metabolome is different between wild-type and cytokinin-deficient plants. Even though it might make the text look excessively long and cumbersome, it is necessary to be clear at this point because these results appear to be of central importance to your manuscript. Please re-write this whole paragraph, and make sure that you:
- first state the differences between wild-type and cytokinin-deficient plants in control conditions;
- then proceed to describe the changes brought about by the exposure to flg22 in wild-type plants;
- then, finally (and most importantly), point out the important differences between the flg22-induced changes in wild-type plants, and the flg22-induced changes in cytokinin-deficient plants. Not the differences between the treated plants; we need to see the differences between the changes that occur in the two genotypes.
- This way you will be able to point out clearly how cytokinin deficiency specifically affects the responses of Arabidopsis plants to the exposure to flg22.
Response 21: We have reorganised this paragraph as your suggestion.
Point 22: Fig. 4: The conceptualization for this figure is excellent, but the figure itself is flawed. As far as I understand the colour scale, the various nuances of red-to-green stand for how much a certain metabolite is altered by the exposure to flg22 in wild-type, or in HvCKX2 plants. However, the dark grey colour stands for "NA" (Not applicable? Not available?) I would not even notice if the "not applicable" colour stood for one or two metabolites, but it stands there virtually for at least one of the genotypes next to every metabolite. Please provide an explanation as to what "NA" stands for, and re-check your data to make the diagram informative as appropriate.
Response 22: “NA” stands for “not affected” and indicates the compounds that have no statistically significant differences between treatments. We have explained it in Figure 4 and Figure 5 captions.
Point 23: line 276: "schematic diagram with heatmaps highlighted metabolic changes" – this figure caption is a total mess. Beside being grammatically incorrect it is abundant with superfluous words (i.e., every diagram is always necessarily schematic). You might want to change this into something like "Heatmap representation of metabolic changes in flg22-responsive..."
Response 23: We have rephrased the caption of Figure 4.
Point 24: line 284-286: This sentence is wrong on at least two levels. It sounds like you are introducing the differences between complete metabolomics of untreated Col0 and HvCKX2 plants, whereas you are actually about to introduce the specific differences in amino acid content, similarly like you did for the sugars in the previous paragraph. Kindly revise.
Response 24: We have rephrased this section and is hopefully clear now.
Point 25: line 287: "plant response", not "plant's response"
Response 25: We have rephrased this as “The metabolic maps of the response to flg22 peptide in HvCKX2-overexpressing plants revealed…”.
Point 26: the entire paragraph 284-297: for the sake of clarity of your results, please try to stick to the same order as for the paragraph 263-271:
- 1) differences between wild-type and cytokinin-deficient plants in control conditions;
- 2) most important changes induced by flg22 in the metabolome of wild-type plants;
- 3) most important differences between what is induced in wild-type plants, and what is induced in cytokinin-deficient plants, by flg22
Response 26: We have reorganised this paragraph as your suggestion.
Point 27: Fig. 5 is far less understandable than Fig. 4 in terms of what is actually shown in the figure. Whereas Fig. 4 (supposedly?) shows how much is each metabolite altered in wild-type, or in HvCKX2, by the exposure to flg22, Fig. 5 shows just only one colour for each metabolite. Why?? What does this colour represent?
- upregulation by flg22 in HvCKX2 plants?
- upregulation by flg22 in control plants?
- the ratio between these two upregulations?
- Please thoroughly revise Fig. 5 and build it on the same logic that you used for Fig. 4.
- The figure caption for Fig. 5 needs to be completely re-written in order to explain what is actually shown in the figure. You should not leave it up to the reader to guess what it is that you wanted to show in the diagram.
Response 27: We have included one more Figure that highlight the changes of CK-sensitive metabolites under untreated conditions to show the effect of flg22 treatment on these metabolites.
Point 28: line 303: You cannot say that the plants "deal" with the peptide. Rephrase to "react to".
Response 28: We have rephrased this following the reviewer’s comment.
Point 29: lines 310-314, 322-325: The abbreviated names of each protein should be fully spelled at first mention.
Response 29: We have included the full name of the proteins.
Point 30: line 326: "several proteins in the biosynthetic pathways..." – please correct to "several enzymes" if that is what you meant
Response 30: We have changed to “several proteins related to the biosynthetic pathways of…”
Point 31: Figure 6: 6b, text within the figure itself – here you are spelling "defense" in American English spelling, whereas you used the British spelling "defence" everywhere else in the manuscript. Please keep the consistency.
Response 31: We have changed ‘defense’ to ‘defence’ through the whole MS.
Point 32: § 6d – Figure caption mentions four subfigures (6a, b, c, d) whereas the figure contains only a, b, and c.
Response 32: This was a mistake in the original caption; there are only three subfigures. It has now been corrected.
Point 33: § Figure caption – revise the figure caption and make it more detailed to point out what is exactly shown in each of the subfigures. Please write a distinct caption for each subfigure. The comment applies to figures 1-5 as well.
Response 33: We have rephrased the figure captions.
Point 34: line 335: Exactly at a point where one would expect the word ("green") in the brackets, suddenly a reference which is missing from the reference list ("Mach et al."), appears in the brackets, apparently for no reason. Please double-check and revise if necessary.
Response 34: This was a mistake in the original caption, and it has now been corrected. It was indeed supposed to read “green”.
Point 35: - Discussion – The results are not properly discussed at all and the entire Discussion section should be re-written from scratch. Among other things, it should be pointed out and discussed whether the observed differences between HvCKX2 and wild-type Arabidopsis plants in response to flg22 can be interpreted as superior or inferior adaptability of HvCKX2 plants as a result of cytokinin deficiency. Appropriate Conclusions to your work should be written accordingly. Detailed comments on the Discussion follow:
o The entire discussion calls for thorough commentary on the interpretation of the adaptive significance of the differentially expressed genes, proteins, and metabolites. Some genes, proteins, or metabolites, that protect the plant tissues against oxidative stress, might be more efficiently upregulated in the cytokinin-deficient plants than in the Col0 plants. Does this necessarily mean that the cytokinin-deficient plants are fitter for the mitigation of stress than the Col0 plants? Or, can it be hypothesised that the protective mechanisms that are more upregulated in HvCKX2 than in Col0, are actually less efficient, "backup" strategies for stress mitigation, that need to be switched on for the plants to respond to stress when their primary stress response mechanisms are insufficient (i.e., due to cytokinin deficiency)?
- Please discuss more on this issue and provide evidence that would support the "greater adaptivity" or "lesser adaptivity" of the cytokinin-deficient plants for the major differences between the HvCKX2 and Col0 plants in response to stress.
- Try to build your entire discussion around the "greater adaptivity" or "lesser adaptivity" paradigm, instead of discussing every trait that you measured one by one, without a logical thread that connects them all.
Response 35: Parts of the discussion have been re-written, and we have now mentioned the possibility of alternative defence responses being favoured in CK-deficient plants. However, the data obtained in the present study does not allow to clearly predict how CK-deficient plants would respond to an actual pathogen infection. Some of the alterations in the defence response could be indicative of a higher resistance (increase in callose deposition, upregulation of certain defence-related proteins), while others suggest a higher susceptibility (decrease in ROS outburst, decrease in SA content). Therefore, we would prefer to avoid making conclusions on the effect of CK-deficiency on plant adaptivity.
Point 36: line 356: the names of the proteins need to be fully spelled at first mention
Response 36: We have fully spelled the protein names.
Point 37: line 388-390: Why is it in contradiction? If cytokinin deficiency leads to lowering the sugar levels in the leaves in normal conditions, does that necessarily mean that the sugars will be unable to accumulate in response to stress? I do not think that these results are contradictory per se.
Response 37: We have rephrased this part of the discussion.
Point 38: line 396-399: Is this a difference from the Col0 plants? If so, please provide brief commentary
Response 38: qP values were lower in mock Col-0 plants, compared mock to HvCKX2 plants, and did not increase after flg22 application. We have now noted this in the text.
Point 39: § line 343-345: This sentence is completely incorrect. The levels of SA are increased in both HvCKX2 and Col0 plants when they are exposed to flg22. Furthermore, the levels of SA and JA are not shown in Figure 1 but in Figure 3. Please read your own text after you write it, and aim for your sentences to be specific, and technically correct. § line 347-348: Whatever you wanted to say in this sentence, is completely incorrect. SA and JA are not "antagonistically regulated".
Response 39: This section of the discussion has been re-written.
Point 40: § line 365: "were affected" – How affected? "+" or "-" ?
Response 40: Both parameters decreased. We have now specified it in the text.
Point 41: § line 366-373: This part of text is written in very confusing manner, and written like this, it definitely "does not show" anything.
Response 41: Parts of this paragraph has been rewritten and new citations added.
Point 42: - Conclusion – please make the Conclusions section separate from the Discussion, and re-write it from scratch after you have thoroughly re-conceived your Discussion around the hypothesis of adaptive or counter-adaptive meaning of the observed differences between HvCKX2 and Col0 plants in response to the flg22 peptide.
Response 42: We have made the Conclusions a separate section and re-written parts of it to reflect the changes in the discussion.
Point 43: - Inconsistent usage of italic letters, capital letters, superscript, and other – Throughout your manuscript, latin names (such as Arabidopsis), gene names (such as HvCKX2), statistically-related terms (such as p-value or Student's t-test), conformation of cytokinins (cis-zeatin and trans-zeatin) which should be written in italic, are left in plain text 9 times out of 10. Furthermore, the names of sugars and amino acids are written randomly in lowercase letters or capitalised, and with or without the designation of molecular conformation (L- or D-) without any criterium whatsoever (e.g., in line 290-292, you write: "L-leucine, L-valine, proline, Methionine and L-isoleucine, alanine, threonine, and L-arginine"). This makes the text of your manuscript look untidy and messy, which is a bit direspectful to the readers. Please take care of that.
Response 43: We have checked the whole manuscript and corrected any inconsistency regarding the usage of italic and capital letters, and the designation of molecular conformation.
Reviewer 2 Report
This manuscript describes the effect of low cytokinins in effector-triggered immune responses in Arabidopsis thaliana.
The authors utilize the CKX2 overexpressing line, which has documented reduced cytokinin content and trigger immune responses using the flg22 peptide.
They screen for early responses like ROS accumulation and callose deposition, as well as for differences in photosynthesis parameters.
Then they use metabolomic and proteomic approaches to elucidate the mechanism.
The manuscript is generally well written and presented without a major issue in the presentation and interpretation of the data.
Some minor issues
-the manuscript would benefit is the authors explained earlier and more extensively that the HvCKX2 line represent cytokinin deficiency.
line 347 it is a little confusing, maybe the authors could elaborate a little more
Author Response
Response to Reviewer 2 Comments
Point 1: - the manuscript would benefit is the authors explained earlier and more extensively that the HvCKX2 line represent cytokinin deficiency.
Response 1: We have included a sentence in the introduction to explain this, and briefly mention it in the abstract.
Point 2: line 347 it is a little confusing, maybe the authors could elaborate a little more
Response 2: This sentence has been removed from the manuscript.
Reviewer 3 Report
This investigation on the influence of cytokinin deficiency on the plant’s most common reactions to the elicitor flg22 comprises a set of well-chosen experiments and evaluation of datasets. Most of the experiments, with the notable exception of figure 1b, were carried out well and were conducted with appropriate methods. The data per se look sound. However, much more information could be retrieved from the data by using more appropriate statistics. The experimental design calls for a two-factor ANOVA as the proper statistical method to assess the interplay between cytokinin deficiency and flg22-elicited pathogen response. I suggest that the authors do so for the experiments shown in Figures 1, 2, and 3 and re-interpret the data if need be.
The legend to figure 1b is incomplete. From the text, I guess that figure 1b describes the state of ROS levels 24 h after flg22 treatment. This information, if guessed correctly, must appear in the legend. Moreover, this experiment should be repeated to collect data at 8 to 12 h after flg22 treatment in order to catch the time point with the largest difference and generate a statistical significance.
Figure 6 d is missing, so the related data cannot be assessed.
Author Response
Response to Reviewer 3 Comments
Point 1: The experimental design calls for a two-factor ANOVA as the proper statistical method to assess the interplay between cytokinin deficiency and flg22-elicited pathogen response. I suggest that the authors do so for the experiments shown in Figures 1, 2, and 3 and re-interpret the data if need be.
Response 1: We have performed both single- and two-factor to interpret the data shown in Figure 1, Figure 2 and Figure 3.
Point 2: The legend to figure 1b is incomplete. From the text, I guess that figure 1b describes the state of ROS levels 24 h after flg22 treatment. This information, if guessed correctly, must appear in the legend.
Response 2: The reviewer is correct; Fig 1b shows ROS levels 24 hours after the treatment. We have now included this in the legend.
Point 3: Moreover, this experiment should be repeated to collect data at 8 to 12 h after flg22 treatment in order to catch the time point with the largest difference and generate a statistical significance.
Response 3: We actually performed pre-experiment at 1h, 12 h, 24h and 48h of flg22 treatment on early defence events such as ROS production and callose deposition in HvCKX2 plants and Col_0 plants, and 24h of flg22 treatment is the time point with the largest difference.
Point 4: Figure 6 d is missing, so the related data cannot be assessed.
Response 4: This was a mistake in the figure footing. There was not supposed to be Figure 6d. We have now corrected this.
Reviewer 4 Report
The manuscript is well-written. Only marginal comments:
line 80: the reference is not mentioned in the list...
line 109: the reference is not mentioned in the list...
line 217,230,234,240,241,242, 285: italics...?
line 335: the reference is not mentioned in the list...

Author Response
Response to Reviewer 4 Comments
Point 1: line 80: the reference is not mentioned in the list...
Response 1: We included this citation in the list.
Point 2: line 109: the reference is not mentioned in the list...
Response 2: We included this citation in the list.
Point 3: line 217,230,234,240,241,242, 285: italics...?
Response 3: We have italicized these terms.
Point 4: line 335: the reference is not mentioned in the list...
Response 4: We have italicized these terms. This was a mistake. The text should have said “(green)”. It has now been corrected.
Round 2
Reviewer 1 Report
Dear Authors,
I was asked for a second round of review for your manuscript "Cytokinin deficiency alters leaf proteome and metabolome during effector-triggered immunity in Arabidopsis thaliana plants", submitted for publication in Plants.
Your manuscript has largely benefitted from the first round of review. The Results are now much more clearly presented both in text and in figures, the Discussion section is mostly useful and meaningful (although parts of it are still flawed), and most of the lettering inconsistencies that were previously scattered throughout the manuscript, have been taken care of. The manuscript is in a much better shape but an additional round of review will be necessary to remove the remaining inconsistencies and to improve the Discussion section, in order for the paper to be recommendable for publication.
- Discussion – although it has substantially improved, Discussion remains the most problematic part of your manuscript. Some parts of Discussion are superfluous and should be deleted, whereas some of your results need to be more properly discussed in the context of cytokinin deficiency and plant defence response. Also, some statements in the Discussion section are incorrect or insufficiently clear:
o line 376: the transgenic plants did not "decrease their SA levels" in response to flg22, they only failed to increase them as intensely as wild-type plants did. Kindly revise.
o line 382-383: both the use of prepositions, and literature citing style in this sentence are unacceptable. Please correct to: "an increase in JA concentration [34] and signalling [35] IN SA-deficient Arabidopsis mutants"
o line 394, 395: "t" and "g" in the protein names need to be in lowercase, A remains in capital case
o line 410-418: The rest of this paragraph, starting from "This seems to be in contradiction..." gives too much detail while leaving the reader without a clear conclusion: Was cytokinin deficiency actually shown to be favourable to callose deposition in the previous studies, or not?!? Your text basically says that it was unfavourable, but not really very unfavourable, actually maybe not unfavourable at all. Please either delete this part, or condense it into a single, preferably not very long sentence.
o line 420: Why TRX5 in particular, and not the other cell wall-related proteins that were identified in your study? Please elaborate.
o line 423: "O" in "flavone 3'-O-methyltransferase" should be written in italic (not necessary in the abbreviated form)
o line 435: What could have caused the decrease in starch synthesis in HvCKX2 plants? This information is out of context and will be unimportant for the discussion if you are unable to put it in the context of cytokinin deficiency and/or plant defence response. More discussion on how HvCKX2 could have affected starch synthesis, is needed here.
o line 440: Again: in the context of cytokinin deficiency and/or defence response: What could be the reason(s) for the repression of photorespiration in HvCKX2 plants? Please provide more discussion on this issue
o lines 441-454: This part of Discussion is robust and very well elaborated. Please try to stick to this standard throughout the Discussion section
o lines 455-459: This paragraph should be deleted because it represents an attempt at discussing the work of other researchers and has no value in the context of your own work
- Introduction
o line 44: Here, after the reference [10], brief commentary about the benefits of using HvCKX2 over the AtCKX genes, should be given.
- Material & Methods – Material & Methods are mostly excellently written, except that unlike in the rest of the manuscript, some words remain in American English spelling whereas the rest of the manuscript is in British spelling, so I would suggest putting these words into British English:
o line 143: analysed
o line 148: normalisation
o line 156: visualising
o line 158: "using sing"??
o line 160: analysed
o line 175: visualisation
- Results
o line 200-201: This sentence does not make sense and needs revision. "The flg22 treatment... may ameliorate the initial ROS outburst... after the flg22 treatment"
o line 218: Are you sure that Figure 1c shows two time points (0h and 24h)? To me it seems that only one time point is shown in Figure 1c, same as for 1b and 1d.
o line 220: What time point is shown in Figure 1d for stomatal aperture index?
o Figure S1: Figure S1 is informative and supports well your findings, thus I would suggest to renumber it as Figure 2 and make it integral part of the paper. If you decide to do this, you should renumber the rest of the figures accordingly, including in the manuscript text.
o line 231: Same comment as for Figure 1c: Figure caption says there are two time points (0h and 24h), but only one time point seems to be shown in the figure.
o line 239: Please delete "(Figure 2a)", you are citing it in the next line as well
o line 306: "uniquely accumulated", or "differentially accumulated"? Please double-check, and revise if necessary.
o line 309-310: "D-mannose" is mentioned twice, kindly revise
o line 312: please delete "and" between L-methionine and L-isoleucine, and replace it with a comma
o line 316-317: I believe that you wanted to say "especially amino acids", not "especially sugars and amino acids"
o Figure 5a: in the manuscript text, you mention that L-alanine was differentially affected, but in Figure 5a, a circle (supposedly greenish in colour) is missing next to L-alanine
- Language
o line 13: please replace "is: To reveal" with "was to reveal", without a colon and without a capital letter
o line 39: "process" should be put to plural ("processes")
o line 50: brackets should be closed after "SA"
o line 207: there is a comma that should be a full stop
o line 267: please correct "than in Col_flg22 plants" to "as in Col_flg22 plants"
o line 287: "amino acids" should be written in plural here
o line 332: "gene expression", not "genes expression"
- The use of lettering and scientific names
o line 32: "N6" should be written as "N6" (N in italic, 6 in plain text but in superscript) both times at line 32
o line 33: "Δ2" should be written as "Δ2", with 2 in superscript
o line 33: t in "tZs" and c in "cZs" should be written in italic letters
o line 36: the currently accepted full name of the CKX enzymes is CYTOKININ OXIDASE/DEHYDROGENASE
o line 37: t in AtCKX7 should be written in lowercase
o line 244: There is no reason for "two-way ANOVA" to be written with a capital T
o line 335, 336: "S" in "glutathione S-transferase" should be written in italic (not necessary in the abbreviated form)
o line 339: "O" in "flavone 3'-O-methyltransferase" should be written in italic (not necessary in the abbreviated form)
o line 341: "t" and "g" in the protein names need to be in lowercase, A remains in capital case
o line 344: GOX1 is glycolate oxidase 1, not glycolate oxidase 2
- Author Contribution Statement
o Please write the Author Contribution Statement in line with CRediT Taxonomy guidelines: https://credit.niso.org/
o Also please make sure that all the Authors are mentioned in the Contribution Statement, as currently, one of the Authors is not mentioned in the Statement.
Author Response
- Point 1:
- line 376: the transgenic plants did not "decrease their SA levels" in response to flg22, they only failed to increase them as intensely as wild-type plants did. Kindly revise.
- line 382-383: both the use of prepositions, and literature citing style in this sentence are unacceptable. Please correct to: "an increase in JA concentration [34] and signalling [35] IN SA-deficient Arabidopsis mutants"
- line 394, 395: "t" and "g" in the protein names need to be in lowercase, A remains in capital case
- line 410-418: The rest of this paragraph, starting from "This seems to be in contradiction..." gives too much detail while leaving the reader without a clear conclusion: Was cytokinin deficiency actually shown to be favourable to callose deposition in the previous studies, or not?!? Your text basically says that it was unfavourable, but not really very unfavourable, actually maybe not unfavourable at all. Please either delete this part, or condense it into a single, preferably not very long sentence.
- Response 1: We have simplified this paragraph to highlight that callose deposition can be achieved under low levels of CK, SA, and ROS outburst.
- Point 2: line 420: Why TRX5 in particular, and not the other cell wall-related proteins that were identified in your study? Please elaborate.
- Response 2: This was a misinterpretation. According to our results, TRX5 is essential for defense response but not be related to callose accumulation. We have removed this mistake from the sentence.
- Point 3: line 423: "O" in "flavone 3'-O-methyltransferase" should be written in italic (not necessary in the abbreviated form)
- Response 3: We have corrected it.
- Point 4: line 435: What could have caused the decrease in starch synthesis in HvCKX2 plants? This information is out of context and will be unimportant for the discussion if you are unable to put it in the context of cytokinin deficiency and/or plant defence response. More discussion on how HvCKX2 could have affected starch synthesis, is needed here.
- Response 4: We have highlighted the role of cytokinin on starch synthesis.
- Point 5: line 440: Again: in the context of cytokinin deficiency and/or defence response: What could be the reason(s) for the repression of photorespiration in HvCKX2 plants? Please provide more discussion on this issue
- Response 5: We have discussed the possibility of the repression of photorespiration in HvCKX2.
- Point 6: lines 455-459: This paragraph should be deleted because it represents an attempt at discussing the work of other researchers and has no value in the context of your own work
- Response 6: We have now deleted this paragraph.
- Point 7: line 44: Here, after the reference [10], brief commentary about the benefits of using HvCKX2 over the AtCKX genes, should be given.
- Response 7: We have now included a brief commentary before that sentence (after reference).
- Point 8:
- line 143: analysed
- line 148: normalisation
- line 156: visualising
- line 158: "using sing"??
- line 160: analysed
- line 175: visualisation
- Response 8: We have put these words into British English.
- Point 9: line 200-201: This sentence does not make sense and needs revision. "The flg22 treatment... may ameliorate the initial ROS outburst... after the flg22 treatment"
- Response 9: We have deleted this sentence.
- Point 10: line 218: Are you sure that Figure 1c shows two time points (0h and 24h)? To me it seems that only one time point is shown in Figure 1c, same as for 1b and 1d.
- Response 10: The reviewer is right; the figures only show data 24 hours after treatment. It has now been corrected.
- Point 11: line 220: What time point is shown in Figure 1d for stomatal aperture index?
- Response 11: 24 hours after treatment. We have now included in the figure caption.
- Point 12: Figure S1: Figure S1 is informative and supports well your findings, thus I would suggest to renumber it as Figure 2 and make it integral part of the paper. If you decide to do this, you should renumber the rest of the figures accordingly, including in the manuscript text.
- Response 12: We agree. We have now made figure S1 a “normal” figure (Figure 2) and renamed the rest.
- Point 13: line 231: Same comment as for Figure 1c: Figure caption says there are two time points (0h and 24h), but only one time point seems to be shown in the figure.
- Response 13: As for Fig. 1c, it only shows 24 hours after treatment. We have now included in the figure caption.
- Point 14: line 239: Please delete "(Figure 2a)", you are citing it in the next line as well.
- Response 14: We have deleted it.
- Point 15: line 306: "uniquely accumulated", or "differentially accumulated"? Please double-check, and revise if necessary.
- Response 15: We have corrected it to “differentially accumulated”.
- Point 16:
- line 309-310: "D-mannose" is mentioned twice, kindly revise.
- line 312: please delete "and" between L-methionine and L-isoleucine, and replace it with a comma
- line 316-317: I believe that you wanted to say "especially amino acids", not "especially sugars and amino acids"
- Response 16: The reviewer is right. We have corrected it.
- Point 17: Figure 5a: in the manuscript text, you mention that L-alanine was differentially affected, but in Figure 5a, a circle (supposedly greenish in colour) is missing next to L-alanine
- Response 17: I have filled the missing.
- Point 18:
- line 13: please replace "is: To reveal" with "was to reveal", without a colon and without a capital letter
- line 39: "process" should be put to plural ("processes")
- line 50: brackets should be closed after "SA"
- line 207: there is a comma that should be a full stop
- line 267: please correct "than in Col_flg22 plants" to "as in Col_flg22 plants"
- line 287: "amino acids" should be written in plural here
- line 332: "gene expression", not "genes expression"
- line 32: "N6" should be written as "N6" (N in italic, 6 in plain text but in superscript) both times at line 32
- line 33: "Δ2" should be written as "Δ2", with 2 in superscript
- line 33: t in "tZs" and c in "cZs" should be written in italic letters
- line 36: the currently accepted full name of the CKX enzymes is CYTOKININ OXIDASE/DEHYDROGENASE
- line 37: t in AtCKX7 should be written in lowercase
- line 244: There is no reason for "two-way ANOVA" to be written with a capital T
- line 335, 336: "S" in "glutathione S-transferase" should be written in italic (not necessary in the abbreviated form)
- line 339: "O" in "flavone 3'-O-methyltransferase" should be written in italic (not necessary in the abbreviated form)
- line 341: "t" and "g" in the protein names need to be in lowercase, A remains in capital case
- line 344: GOX1 is glycolate oxidase 1, not glycolate oxidase 2
- Response 18: We have corrected all of them.
- Point 19: Please write the Author Contribution Statement in line with CRediT Taxonomy guidelines: https://credit.niso.org/
- Response 19: Author contribution have been rewritten based on CRediT Taxonomy guidelines.
- Point 20: Also please make sure that all the Authors are mentioned in the Contribution Statement, as currently, one of the Authors is not mentioned in the Statement.
- Response 20: We have mentioned all authors for their contribution.